**Data Availability Statement:** Data available in the manuscript and Supporting Information files.

**Funding:** The authors received no specific funding for this work.

# Spatial distribution of suspected and confirmed cholera cases in Mwanza City, Northern Tanzania

**Monica T. Madullu[1], Deborah S. K. Thomas[2], Elias C. Nyanza[1]\*, Jeremiah Seni[3], Sospatro E. Ngallaba[1], Sophia Kiluvia[4], Moses Asori[2], Joseph Kangmennaang[5]**

**1** Department of Environmental, Occupational and Research GIS, School of Public Health, Catholic University of Health and Allied Sciences, Mwanza, Tanzania, **2** Department of Geography and Earth Sciences, University of North Carolina at Charlotte, Charlotte, North Carolina, United States of America, **3** Department of Microbiology and Immunology, Weill Bugando School of Medicine, Catholic University of Health and Allied Sciences, Mwanza, Tanzania, **4** Environmental Health Department, Mwanza City Council, Mwanza, Tanzania, **5** School of Kinesiology and Health Studies, Queens University, Kingston Ontario, Canada

\* elcnyanza@gmail.com

## Abstract

Cholera, which is caused by *Vibrio cholerae*, persists as a devastating acute diarrheal disease. Despite availability of information on socio-cultural, agent and hosts risk factors, the disease continues to claim lives of people in Tanzania. The present study explores spatial patterns of cholera cases during a 2015–16 outbreak in Mwanza, Tanzania using a geographical information system (GIS) to identify concentrations of cholera cases. This cross-sectional study was conducted in Ilemela and Nyamagana Districts, Mwanza City. The two-phase data collection included: 1) retrospectively reviewing and capturing 852 suspected cholera cases from clinical files during the outbreak between August, 2015, and April, 2016, and 2) mapping of residence of suspected and confirmed cholera cases using global positioning systems (GPS). A majority of cholera patients were from Ilemela District (546, 64.1%), were males (506, 59.4%) and their median age was 27 (19–36) years. Of the 452 (55.1%) laboratory tests, 352 (77.9%) were confirmed to have *Vibrio cholerae* infection. Seven patients (0.80%) died. Cholera cases clustered in certain areas of Mwanza City. Sangabuye, Bugogwa and Igoma Wards had the largest number of confirmed cholera cases, while Luchelele Ward had no reported cholera cases. Concentrations may reflect health-seeking behavior as much as disease distribution. Topographical terrain, untreated water, physical and built environment, and health-seeking behaviors play a role in cholera epidemic in Mwanza City. The spatial analysis suggests patterns of health-seeking behavior more than patterns of disease. Maps similar to those generated in this study would be an important future resource for identifying an impending cholera outbreak in real-time to coordinate community members, community leaders and health personnel for guiding targeted education, outreach, and interventions.

**Competing interests:** The authors have declared that no competing interests exist.

**Abbreviations:** GIS, Geographic Information System; CREC, Catholic University of Health and Allied Sciences and Bugando Medical Centre; GLLAMM, The Generalized Linear Latent and Mixed Models; GPS, Global Positioning System; WASH, Water, Sanitation, and Hygiene.

## 1. Introduction

Cholera, caused by *Vibrio cholerae*, persists as a devastating acute diarrheal disease and public health challenge in low and middle-income countries [LMICs] [1, 2]. Worldwide cholera case estimates range from 1.3 to 4.0 million, causing between 21,000 and 143,000 deaths every year [1, 3]. The first ten cholera cases in Tanzania were reported in 1974 [4], and the number of cases has continued to increase, frequently causing outbreaks that require taking measures for containment of the disease. In 2006, Tanzania experienced a cholera outbreak with more than 14,297 reported cases and 254 deaths (Case Fatality Rate of 1.8%) [5]. Again, in January 2015 to January 2018, 33,421 cases, including 542 deaths were reported, indicating persistent challenges with cholera despite various interventions [6]. Mwanza, located in northwestern part of Tanzania, had more than 12.6% of the reported cases [7]. However, numbers are likely underreported, as has been noted in other countries due to political situations, health-seeking behavior, cultural beliefs, stigmatization surrounding the disease and weak laboratory capacity [3, 8, 9].

Environmental, socio-cultural and economic factors contribute to primary and secondary cholera transmission [10–14]. Areas with limited access to safe drinking water and good sanitation are susceptible to cholera outbreaks [1]. Fecal oral transmission, which is the most common means of acquiring the diseases, occurs because of poor sanitation and hygiene practices [1, 3]. Tanzania continues to invest in health promotion strategies towards improving water, sanitation, and hygiene (WASH) in efforts to fight cholera, but outbreaks persist [14, 15].

Cholera pathogenesis and associated risk factors that pre-dispose human population to cholera are well documented. Even so, cholera outbreaks continue reemerging year after year at an alarming rate [16], contributing significantly to the morbidity and mortality throughout the world [17]. As such, the need for measures to prevent reoccurrence of the disease by the delineation of cholera hotspots to target specific preventive measures. Recently, there has been a growing concern that purely individual-based analyses of the cause of disease are insufficient and fail to capture important disease determinants [18].

In Tanzania, once a person is suspected of having cholera at a health facility, s/he is isolated in a special room for treatment and observation. If a sudden increase in the number of cholera cases that are linked by time and place occurs, health authorities declare an outbreak. When a cholera outbreak is declared, suspected cholera cases (any individual in any age group who experience three or more acute watery diarrhea stools in 24-hours, with or without vomiting) are referred to isolated camps as a containment measure to have designated locations for cholera care and to minimize spread of the disease. Patients are assessed, treated, and discharged once they do not show any indication of the disease. Transfer to the camp is considered mandatory and all people suspected of having cholera are expected to comply. In this regard, the current study focused on understanding the spatial distribution of suspected cholera cases in Ilemela and Nyamagana municipals.

Geographic information systems (GIS) is frequently applied as a tool for exploring spatial patterns and relationships in public health, including examining cholera [19]. Studies have identified clusters of cholera disease and evaluated an array of risk factors across geographic areas in various parts of the world [20–23]. Using "big data" techniques, Lessler et al. [23] draw on numerous spatial datasets across sub-Saharan Africa to portray current patterns to guide targeted control efforts. In Tanzania, researchers utilized GIS identified sources of cholera in informal urban settlements in Dar es Salaam [10]. This study explores spatial patterns of cholera cases during a 2015–16 outbreak in Mwanza, Tanzania using GIS to identify concentrations of cholera cases, illustrating the potential for GIS applications in more resource-limited settings for monitoring and examining a cholera outbreak.

## 2. Methods

### 2.1. Study design, settings and population

This cross-sectional analytical study was carried out in Ilemela and Nyamagana municipals that make-up Mwanza City located along the southern shores of Lake Victoria in northern Tanzania (Fig 1). Each of these municipals comprised of 19 and 18 wards, respectively. The

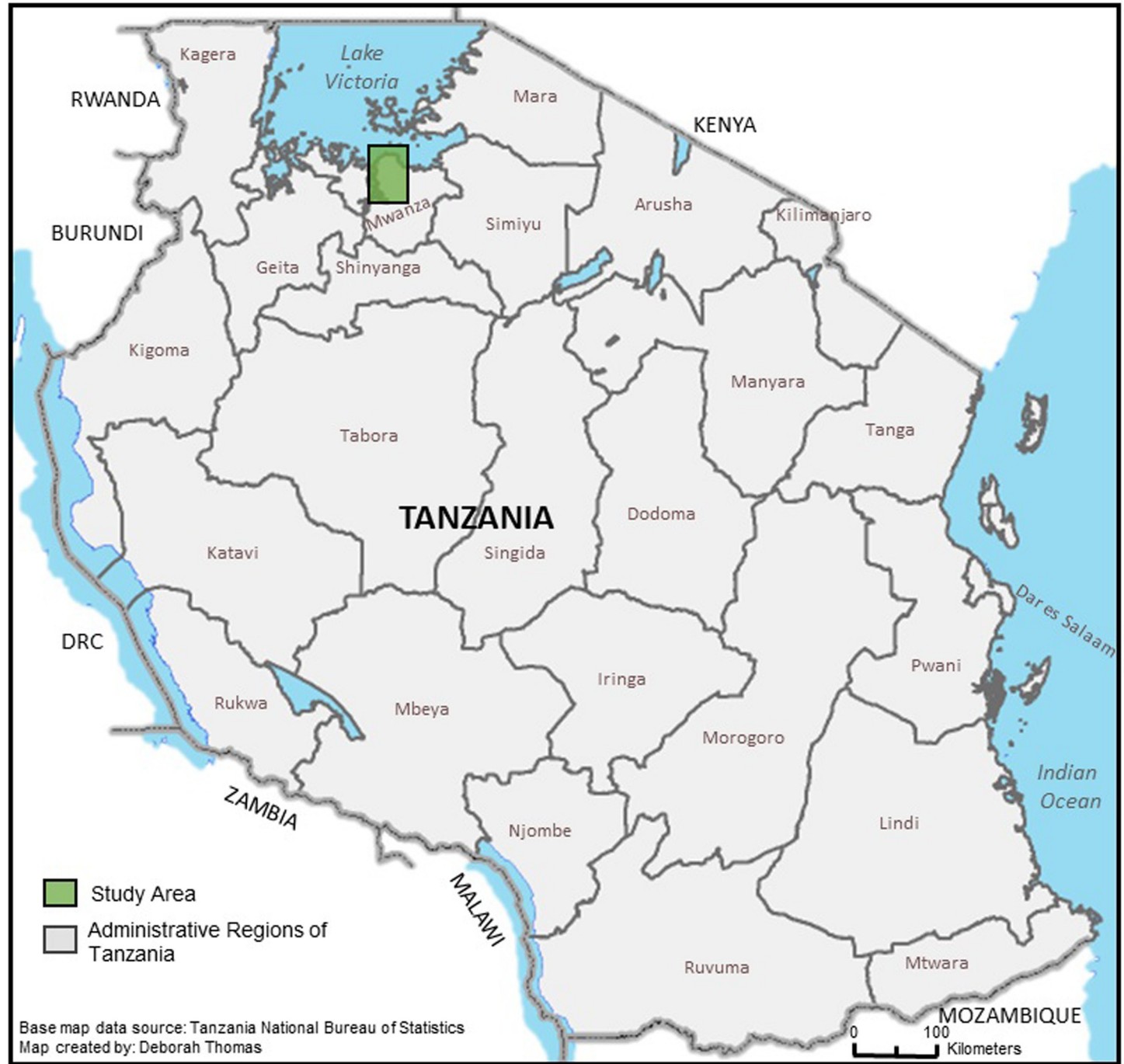

**Fig 1. Study area, Mwanza City, Tanzania.** Base map data source: OCHA, https://data.humdata.org/dataset/cod-ab-tza. Map created by: Deborah Thomas.

city has a total population of 706,453 [24]. In 2015–16 during the outbreak, there were a total of five camps, four permanent ones (Igoma, Karume, Pasiansi, Sangabuye) and one temporary at Bezi Island.

## 2.2. Eligibility criteria

All patients with suspected and/or confirmed with cholera of all age groups and sex who reported to any of the five cholera camps from August 2015, through April 2016, were included in the study (n = 852). Patients' files with no specifications of residence were excluded (n = 3, 0.35%), as were those with a residence outside of Mwanza City (n = 23, 2.6%).

## 2.3. Data collection

The data collection occurred in two phases. Phase I focused on the retrospective collection of secondary data from patients' files from each of the camps from August 2015, through April 2016. The data included: patients' age, sex, occupation, area of residence, water sources, duration spent at the camp, time from symptoms to transfer to the camp, laboratory testing, and outcome.

Phase II mapped the residences of cholera cases during fieldwork conducted May-June, 2016. Because no addresses or street indexing exists in this setting, capturing the locational data involved visiting specific areas of residence for the cholera cases and capturing the global positioning coordinates (GPS) of the household. Identification of the respective patient's household location involved a series of steps. First, the name of the street from where the patient came was recorded from the cholera camp register. Next, having identified the name of the street, one of the research assistants would list all the names from the respective street. Then, the list of names was submitted to the government office where the chairperson of the street in support of the ward leaders identified the households listed from the camp. Lastly, the location of each household was captured using GPS. Data/information collected was anonymously stored and secured on a password protected computer and files and were only accessible to the research team. Each entry was assigned an anonymous code for further protection.

## 2.4. Geo-spatial analysis of cholera cases

Maps were produced using *ArcGIS Desktop 10.3* [25]. Base maps and population numbers were obtained from the Tanzania National Bureau of Statistics [24, 26]. The distance from the household of each case to the respective camp was calculated using Euclidian distance as an estimate of how far people had to travel to the camp. This is a reasonable option given the lack of complete digital street files and the use of walking paths not captured in digital files, such as *OpenStreetMap* [27]. Ripley's *K* [28, 29], which is commonly applied for univariate point pattern detection to assess clusters over a range of distances, rather than a defined distance, was calculated using *ArcGIS Desktop 10.3* software [25] with 999 permutations and 20 distance bands. Kernel density estimation of case home locations was calculated using a three-kilometer search radius, selected as a walkable distance threshold. Rate ratios were calculated for each ward by dividing the percentage of the population in the ward with suspected cholera cases and then dividing by the overall district percentage of the population estimated of having cholera.

## 2.5. Statistical data analysis

The data were also analyzed with a binary logistic regression with a log-link function that assumed a symmetrical distribution to produce the parameter estimates. The outcome variable

(time to seeking care) is binary and almost evenly distributed. The multivariate analysis controlled for variables including distance to health facility, age, gender, occupation, water source. The Generalized Linear Latent and Mixed Models (GLLAMM) program available in *Stata 15* [30] was used to build all models.

### 2.6. Ethical consideration

Ethical approval for the research was obtained from the joint research and ethics review committee of Catholic University of Health and Allied Sciences and Bugando Medical Centre (CREC# 129/2016). Permission to conduct the research was obtained from the relevant authorities at Region and District levels in Mwanza. This was a secondary data analysis with no direct contact with patients.

## 3. Results

### 3.1. Baseline characteristics of the cholera cases

The study included a total of 852 cholera cases in Ilemela and Nyamagana Districts admitted to one of the five cholera camps. Briefly, in Ilemela District there were four camps (*Viz.*, Igoma (n = 298, 34.98%), Pasiansi (n = 165, 19.37%), Sangabuye (n = 148, 17.37%) and Bezi (n = 14, 1.64%), whereas in Nyamagana District there was only one camp (*Viz.*, Karume (n = 227, 26.64%). Few people died (n = 7, 0.80%) and only two patients were referred to a hospital. Table 1 provides a summary of each of the variables from the camp records, including the calculated average distance traveled to the camp from the household. Most cases were from Ilemela District (n = 546, 64.08%). The median age of the patients admitted in cholera camps was 27 (19–36) years old. Most were male (506, 59.39%), and nearly half reported using tap water (n = 419, 49.18%). Seven patients (0.82%) died. Of the adults, most were involved in fishing or farming as an occupation. Almost half of suspected cases reported to a health facility within 12 hours of exhibiting symptoms. Only a little over half of the people at the camps had a laboratory test (n = 452), and 352 tested positive.

Of the 352 confirmed cholera cases, a majority were between 19 and 60 years (n = 248, 70.45%) and were male (n = 220, 62.50%) (Table 2). Six of the seven patients who died were confirmed cases. Individuals who identified as peasants (n = 73, 33.03%) saw the highest rate of confirmed cases among workers, while those in the fishing industry (n = 50, 22.62%) saw the second highest rate of confirmed cases. Among the 352 confirmed cholera cases, diarrhea presented as the most common symptom (n = 348, 98.86%) followed by vomiting (n = 326, 92.61%) and dehydration (n = 314, 89.20%) (Table 3).

### 3.2. Spatial pattern of cholera cases in Mwanza City

Fig 2 displays the geographic pattern of suspected cholera cases across Mwanza City and reveals distinct clustering. Bezi Island, in Lake Victoria on the north and representing only 14 suspected cases is not included on the maps or the clustering statistic. The highest density of suspected cholera case households generally surrounds the camps, with one exception in the central/western area, which has higher population density unplanned settlements with limited sanitation. Ripley's *K* results show significant clustering of cases to 5.7 kilometers.

Sangabuye (6.79) and Bugogwa (4.80) Wards in the northern part area had the highest case rate ratios. Mkolani Ward (0.08) in the southern end and Nyakato Ward (0.15) in the center had the lowest case rate ratios (Fig 3). The higher rates of suspected cholera cases correspond with areas surrounding Sangabuye and Karume Camps.

**Table 1. Characteristics of suspected cholera cases.**

| Characteristics | Total N = 852 N (%) | Average Distance Traveled (km) |
|---|---|---|
| **District** | | |
| Ilemela | 546 (64.08) | 3.28 |
| Nyamagana | 306 (35.92) | 5.87 |
| **Age of participants (years)** | | |
| 0–5 | 90 (10.56) | 5.07 |
| 6–17 | 108 (12.68) | 4.04 |
| 18–60 | 619 (72.65) | 4.09 |
| > 60 | 35 (4.11) | 4.56 |
| **Sex** | | |
| Male | 506 (59.39) | 4.24 |
| Female | 346 (40.61) | 4.16 |
| **Water source** | | |
| Well | 193 (22.65) | 3.69 |
| Lake | 226 (26.53) | 3.23 |
| Tap | 419 (49.18) | 5.03 |
| Others | 2 (0.23) | 4.84 |
| No Response | 12 (1.41) | 2.03 |
| **Occupation** | | |
| Business | 70 (8.22) | 4.20 |
| Driver | 56 (6.57) | 3.61 |
| Fishing | 144 (16.90) | 3.60 |
| Food Service | 61 (7.16) | 3.25 |
| Peasant | 152 (17.84) | 4.38 |
| Service | 66 (7.75) | 4.94 |
| Housewife | 73 (8.57) | 4.59 |
| Student | 31 (3.64) | 3.80 |
| Child/not working | 171 (20.07) | 4.62 |
| Other | 28 (3.29) | 5.00 |
| **Hours until came to camp** | | |
| <12 | 421 (49.41) | 3.77 |
| 12–24 | 339 (39.79) | 4.34 |
| >24 | 92 (10.80) | 5.73 |
| **Laboratory Tested** | | |
| No | 400 (46.95) | 3.92 |
| Yes | 452 (53.05) | 4.47 |
| Positive | 352 (77.88 of tested) | 4.38 |
| Negative | 100 (22.12 of tested) | 4.79 |

## 3.3. Association between baseline characteristics and cholera cases

The multivariate results show that source of drinking water, occupation and age were the most significant determinants of the time suspected or confirmed cases of cholera were reported at a health post. Respondents who indicated wells as their water source (OR = 2.10, $p \leq 0.01$) were more likely to report early (within 12hours) to the hospital compared to those relying on tap water. Compared to business owners, peasant farmers (OR = 2.31, $p \leq 0.05$) were more likely to report to the health facility within the first 12 hours. The only demographic variables associated with health seeking behaviours was age. Respondents age 6 to 17 years (OR = 0.49, $p \leq 0.01$) were less likely to seek care compared to respondents between 0 to 5years.

**Table 2. Laboratory test results for cholera of those tested at the camps.**

| Characteristics | Positive N (%) | Negative N (%) | Total Tested N |
|---|---|---|---|
| **Age (years)** | | | |
| 1–5 | 37 (75.51) | 12 (24.49) | 90 |
| 6–17 | 47 (75.81) | 16 (24.19) | 108 |
| 18–60 | 253 (78.09) | 70 (21.91) | 619 |
| > 60 | 15 (88.24) | 2 (11.76) | 35 |
| **Sex** | | | |
| Male | 220 (78.85) | 59 (21.15) | 279 |
| Female | 132 (76.30) | 41 (23.70) | 173 |
| **Water sources** | | | |
| Well | 78 (82.98) | 16 (17.02) | 94 |
| Lake | 73 (85.88) | 12 (14.12) | 85 |
| Tap | 195 (73.58) | 70 (26.42) | 265 |
| Other | 1 (50.00) | 1 (50.00) | 2 |
| No response | 5 (83.33) | 1 (16.67) | 6 |
| **Hours at home** | | | |
| < 12 hours | 149 (78.84) | 40 (21.16) | 189 |
| 12–24 hours | 155 (78.28) | 43 (21.72) | 198 |
| > 24 hours | 48 (73.85) | 17 (26.15) | 65 |

## 4. Discussion

Cholera persists as a public health challenge in Mwanza City with reemerging cholera outbreaks over time. The present study provides an example of how mapping can be utilized to guide future interventions to reduce cholera epidemics in Ilemela and Nyamagana Districts. Our findings suggest patterns of health-seeking behavior, which is relevant for targeting education and outreach. The disease distribution based on age warrants significant economic concerns, as higher confirmed cases were among the working age population (19–60 years), which is consistence with other studies from other African countries such as Nigeria [31] and Uganda [32]. This has considerable implications on decreased economic productivity and increased dependency burden impacting poverty. Nyamagana and Ilemela Wards were among the areas with the highest number and concentration of cholera cases, both bordering Lake Victoria, which is consistent with a study in Kenya whereby districts bordering large water bodies had 5.5 times increased likelihood of cholera cases [33].

The northern part of the study area near Lake Victoria had an increased burden of cholera cases, particularly in the fishing communities of Sangabuye and Bugogwa. Mabatini and Igoma, which do not border a major water body, exhibited variable distribution of cholera cases. The relative decrease in cholera cases moving away from a water body was reported in another study in Haiti along the Artibonite River and Lake Tanganyika in Burundi [9, 34].

**Table 3. Symptoms of cholera positive and negative patients.**

| Symptom | Positive Patients N = 352 | Negative Patients N = 100 |
|---|---|---|
| | n (%) | n (%) |
| Vomiting | 326 (92.61) | 88 (88.00) |
| Diarrhea | 348 (98.86) | 99 (99.00) |
| Abdominal Cramping | 50 (14.20) | 14 (14.00) |
| Dehydration | 314 (89.20) | 93 (93.00) |

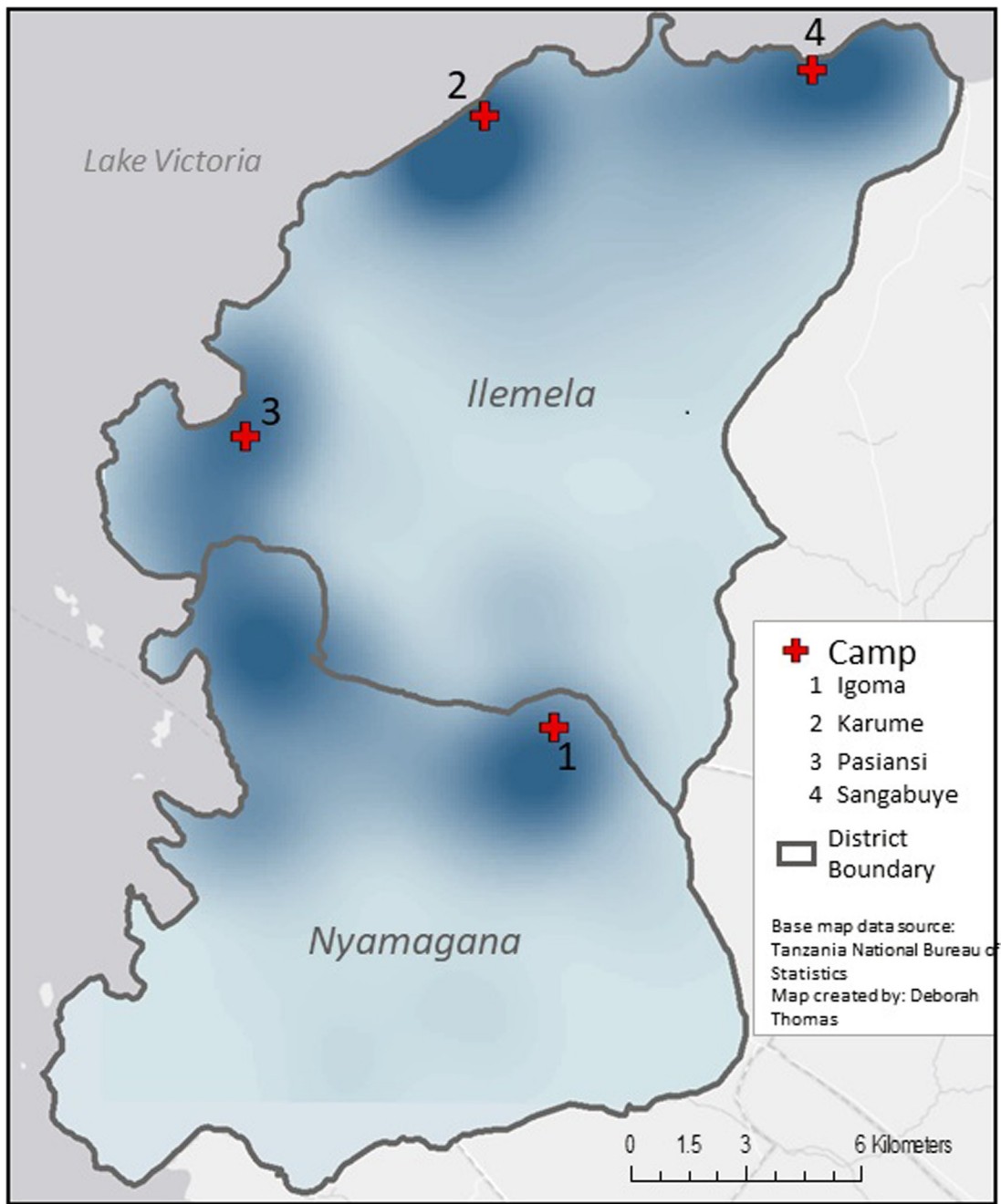

**Fig 2. Kernel density estimation of home locations for suspected cholera cases in Mwanza City, Tanzania, during the August 2015- April 2016 outbreak.** The darker the color, the higher the concentration of where patients at camps lived. Base map data source: OCHA, https://data.humdata.org/dataset/cod-ab-tza. Map created by: Deborah Thomas.

Our study found increased risk among well and tap water users as compared to those who relied on other sources of water. This suggests a need to investigate the interaction between imposed surface water contamination (primary moderated by human activities) and under-ground hydrogeochemistry (based on the nature of aquifers) along with well stewardship, health seeking behavior and socioeconomic conditions to determine differential risk exposures [35, 36].

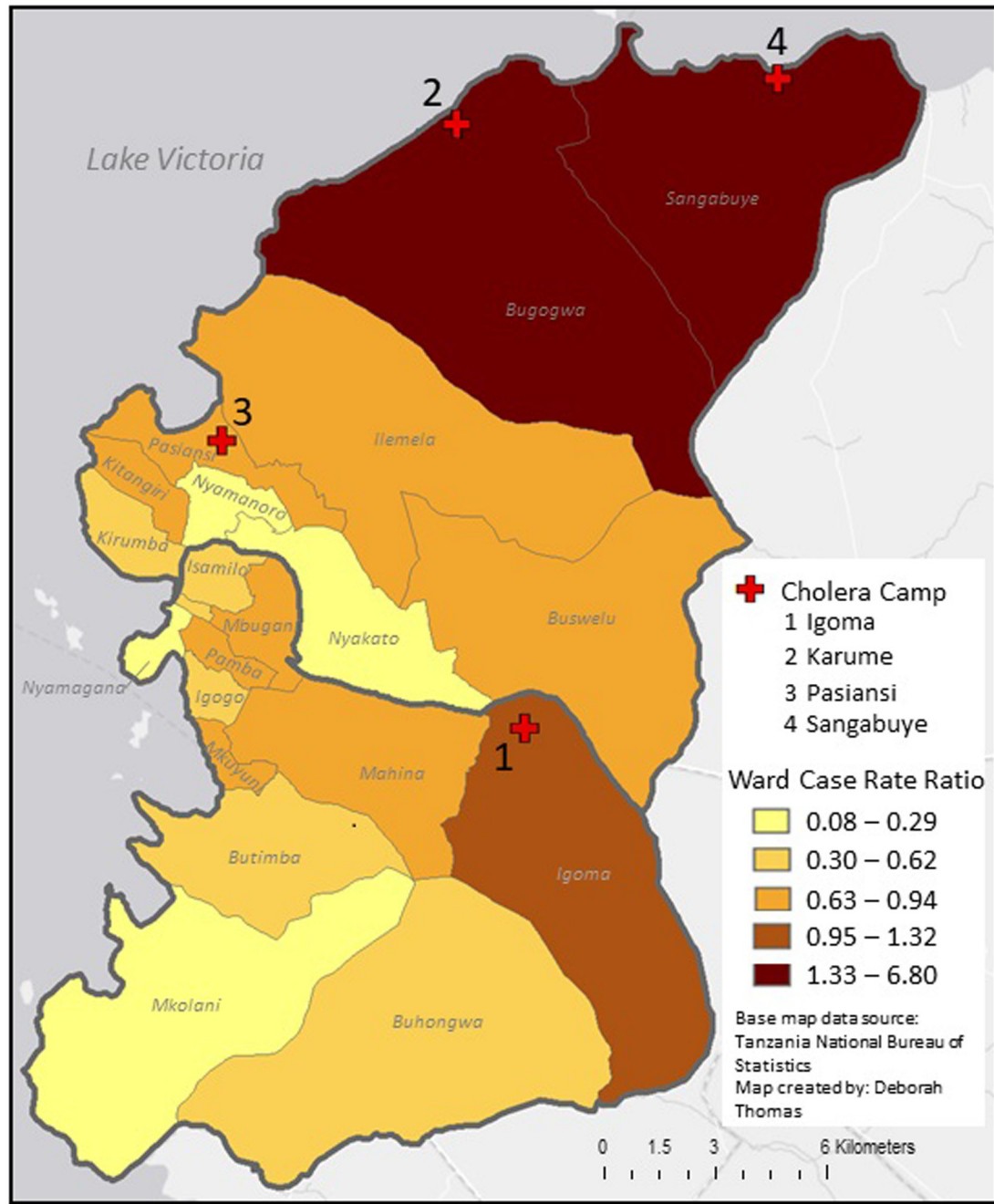

**Fig 3. Case ratio of ward as compared to the case ratio of Mwanza City for patients attending cholera camps for treatment during the August 2015- April 2016 outbreak.** Base map data source: OCHA, https://data.humdata.org/dataset/cod-ab-tza. Map created by: Deborah Thomas.

Interestingly, Luchelele Ward that borders Lake Victoria had neither a suspected or confirmed cholera case reported. This might be attributed to a relatively level and even topography, rather than hilly, which favors construction of permanent structures for human waste disposal. In Mabatini Ward, on the other hand, people live in high hills with hard rocks where it is challenging to build latrines. A spatial analysis of cholera epidemic in Harare, Zimbabwe also reported an association between higher elevation and lower cholera risk [21]. The physical

and built environment plays an important role along with behavioral and social considerations in considering prevention and response to cholera. Related to occupational type, the study found that peasant farmers as compared to those in other occupations (Table 4) had higher odds of reporting early to the treatment centers, highlighting the need to strategically incorporate occupational factor during community education and campaigns.

A recent study in Burundi confirmed that people that resided near the Lake Tanganyika were at higher risk of cholera due to the Lake' role in moderating fishing activities and migration patterns [9]. Residing near major water bodies in the absence of sanitation measures is associated with increased chances of contaminated water sources leading to transmission of

**Table 4. Determinants of time of seeking health care during a cholera outbreak.**

| Independent variables | OR(95%CI) |
|---|---|
| Distance to cholera camp (ref: less than 5km) | |
| >5km | 0.74(0.53–1.03) |
| Water source (ref: tap) | |
| Well | 2.10(1.25–3.51)** |
| Lake | 1.54(0.86–2.74) |
| No Response | 1.85(0.55–6.16) |
| Occupation (ref: Business) | |
| Driver | 1.26(0.38–4.15) |
| Fishing | 1.15(0.53–2.51) |
| Food Service | 1.85(0.92–3.69) |
| Peasant | 2.31(1.05–5.04)* |
| Service | 1.95(0.93–4.06) |
| Housewife | 0.87(0.32–2.31) |
| Student | 1.22(0.63–2.31) |
| Other | 1.95(0.91–4.15) |
| Driver | 2.69(0.96–7.58) |
| Age in categories (ref: 0–5) | |
| 6–17 | 0.49(0.26–0.94)* |
| 18–60 | 0.38(0.12–1.21) |
| > 60 | 0.44(0.11–1.73) |
| Sex (ref: male) | |
| Female | 0.79(0.55–1.13) |
| Camp (ref: Bezi camp) | |
| Igoma HC | 0.66(0.12–3.55) |
| Karume HC | 0.94(0.29–3.01) |
| Pasiansi Disp | 1.51(0.43–5.26) |
| Sangabuye HC | 0.57(0.17–1.85) |
| District of residence (ref: Ilemela) | |
| Nyamagana | 0.71(0.21–2.33) |
| *Random effects* | |
| Ward level | 1.00(0.82–1.21) |
| Constant | 1.91(0.334–10.89) |
| Observations | 852 |

Notes: OR = odds ratio; Ref: = Reference Categories;

*p ≤ .05,

**p ≤ .01; CI = confidence intervals.

water borne diseases. In a parallel fashion, Mwanza City had significant numbers of cholera cases among individuals using tap water, which may be attributable to lack of personal hygiene resulting in transmission of the disease among members within the family and to the near-by residents. Of the 852 individuals admitted in cholera camps nearly three quarters were adults and the majority were male, similar to a study in Lusaka, Zambia and Bangladesh [37, 38]. This perhaps results from the involvement of men in outdoor activities and occupations that exhibit movement across greater areas, exposing them to risks related to contaminated food or water and unsanitary environments.

More than three quarters of the confirmed cholera cases went to the health facilities within 24 hours from the onset of acute diarrhoea, suggesting good health seeking behavior among a subset of city residents. However, 10% attended more than 24 hours from the onset of symptoms. This may be explained by limited access to health facilities attributable to long distance and limited awareness of the disease as shown by other studies in Kenya [11, 33]. Perhaps more than revealing disease clusters during the 2015–16 outbreak, the spatial analysis suggests patterns of health-seeking behavior. The one 'hot spot' exception in the western part of the study area spans from Mkuyuni to Bugarika to parts of Mabatini, which is a densely populated area with limited sanitation and mostly comprised of unplanned settlements. For all other highly concentrated areas of households, suspected cases came from catchment areas closest to the camps; each camp exhibited a concentration of suspected and confirmed cases coming to the camp from within five kilometers. The average time taken for people to seek care reinforces an interpretation of health-seeking behavior for those areas around the camps. As the time increases to diagnosis at a health facility, the average distance also increases, suggesting people wait to seek care if further away, which is in line with other studies [39, 40].

Frequently, once initial cases are confirmed, only representative samples, not all cases, are evaluated using laboratory tests in subsequent suspected cases with the remaining treated with a clinical diagnosis. The present study had approximately half of the cholera suspected cases confirmed by laboratory tests, which is similar to other studies in Kenya and Haiti [33, 34]. This is a normal practice during an outbreak especially for cholera according to the Ministry of Health guideline for cholera outbreak control [41]. Thus, patients' histories and clinical examination provide a substantial portion of data. This could be contributing to overestimation of cholera cases due to recording of individuals with self-reported diarrhea. A recent study in Zanzibar found that most healthcare centers had weak laboratory capacity which significantly hampered the detection and confirmation of cases and outbreaks of Cholera [42]. In this case, such a practice could be underestimating of cholera cases among those who present at a health facility.

Referring confirmed and suspected cases from health facilities to camps for containment and treatment has merit in terms of ensuring care and quarantining those with disease. However, the approach may simultaneously create barriers to care. For example, patients often opt for self-medication at home, avoiding health facilities all together in the efforts to abscond camp containment. Although treatment at the facility is free, containment poses hardships to sustaining livelihoods as well; for example, if spending time at a camp—income is lost, or childcare can be problematic. Thus, active community outreach, education, and interventions must occur in parallel to efforts at health facilities.

## 4.1 Recommendations

While mapping cholera retrospectively in Mwanza City in a cross-sectional manner to understand the outbreak, the approach reveals opportunities for conducting closer to real-time mapping in support of disease management. With the emergence of mobile health (mHealth) data

collection and mapping tools, geographical evaluation and analysis can guide targeted interventions, risk assessments, and importantly educational outreach [41, 42]. Prioritizing the use and allocation of limited resources through the timely identification of a potential impending cholera outbreak can provide guidance for prompt responsive intervention measures. The generated information can be used as a baseline platform for connecting individuals in the risky communities with community leaders and ultimately to health personnel in the respective health care facilities. This collaboration is critical for further exploring the socio-demographical, cultural and clinical determinants of this fatal disease in real-time for minimizing cholera outbreaks.

Since the current findings suggest a significant association between water source and confirmed cholera cases, there is high need to ensure reliable supply of treated water throughout Nyamagana and Ilemela Municipals. It is also important to consider high risky occupational groups, such as public transport operators and fishing communities.

### 4.2 Limitations

Since the present study used secondary data there was no possibility for validation and/or correcting the information provided in the patients' files. A patient's movement, eating, and sanitation habits were not evaluated since they were not part of the secondary data source. Thus, a comprehensive set of risk factors could not be explored. The distribution of point locations do not represent all cases or even a random sample and so the potential for exploring ecological risk factors is limited.

### 5. Conclusions

Topographical terrain, untreated water, physical and built environment, and health-seeking behaviors play a role in cholera epidemic in the study area. The mapping appeared to reveal some clustering of disease, especially suggesting patterns of health-seeking behavior. The study illustrates the use of GIS for identifying an impending cholera outbreak in real-time and subsequently guide targeted links between community members, community leaders and health personnel interventions.

### Supporting information

**S1 Data. Suspected and confirmed cholera cases data.**
(XLSX)

### Acknowledgments

The authors thank Michael Modzelewski for his assistance with generating and checking the data summaries and also Michael Desjardins for his statistical advice.

### Author Contributions

**Conceptualization:** Monica T. Madullu, Deborah S. K. Thomas, Sospatro E. Ngallaba.

**Data curation:** Monica T. Madullu, Jeremiah Seni, Sophia Kiluvia.

**Formal analysis:** Deborah S. K. Thomas, Joseph Kangmennaang.

**Methodology:** Monica T. Madullu, Deborah S. K. Thomas, Elias C. Nyanza, Jeremiah Seni.

**Supervision:** Monica T. Madullu, Deborah S. K. Thomas, Elias C. Nyanza, Jeremiah Seni, Sophia Kiluvia.

**Writing – original draft:** Monica T. Madullu, Deborah S. K. Thomas, Elias C. Nyanza, Jeremiah Seni.

**Writing – review & editing:** Deborah S. K. Thomas, Sospatro E. Ngallaba, Moses Asori, Joseph Kangmennaang.

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
