## [Decision Letter · Decision Letter 0]

7 Sep 2022

PGPH-D-22-01133

Spatial distribution of suspected and confirmed cholera cases in Mwanza City, Northern Tanzania

Dear Dr. Nyanza,

Thank you for submitting your manuscript to PLOS Global Public Health. After careful consideration, we feel that it has merit but does not fully meet PLOS Global Public Health’s publication criteria as it currently stands. Therefore, we invite you to submit a revised version of the manuscript that addresses the points raised during the review process.

We look forward to receiving your revised manuscript.

Kind regards,

David Musoke, PhD

Academic Editor

Journal Requirements:

1. In the online submission form, you indicated that "Aggregated, de-identified data can be obtained by request to the corresponding author.". All PLOS journals now require all data underlying the findings described in their manuscript to be freely available to other researchers, either 1. In a public repository, 2. Within the manuscript itself, or 3. Uploaded as supplementary information.

2. Figure 1, 2 & 3: please (a) provide a direct link to the base layer of the map used and ensure this is also included in the figure legend; (b) provide a link to the terms of use / license information for the base layer. We cannot publish proprietary or copyrighted maps (e.g. Google Maps, Mapquest) and the terms of use for your map base layer must be compatible with our CC-BY 4.0 license.

Please include your response in the response to reviewers file regarding the above concern or include what you have emailed us previously.

3. Please remove the embedded figures from the manuscript file.

4. We have noticed that you have uploaded Supporting Information files, but you have not included a list of legends. Please add a full list of legends for your Supporting Information files after the references list. 

Additional Editor Comments (if provided):

In addition to the reviewer comments:

- A stronger take-away message from the study is needed in the abstract (conclusion).

- In the introduction, there is need to include more details on the knowledge gap that existed (based on available literature on the subject in Tanzania) to justify the need for the study.

- An introductory paragraph is needed in the discussion summarising the main study findings and giving their broad significance to public health.

- Avoid brief discussion paragraphs (see page 10, starting on line 221).

- The conclusion needs to be strengthened. Currently, it is majorly a summary of the study findings. What are the implications of the study findings to policy, practice and programming (if at all)?

Reviewers' comments:

Reviewer's Responses to Questions

**Comments to the Author**

1. Does this manuscript meet PLOS Global Public Health’s publication criteria? Is the manuscript technically sound, and do the data support the conclusions? The manuscript must describe methodologically and ethically rigorous research with conclusions that are appropriately drawn based on the data presented.

Reviewer #1: Yes

Reviewer #2: Yes

2. Has the statistical analysis been performed appropriately and rigorously?

Reviewer #1: Yes

Reviewer #2: Yes

3. Have the authors made all data underlying the findings in their manuscript fully available (please refer to the Data Availability Statement at the start of the manuscript PDF file)?

Reviewer #1: Yes

Reviewer #2: Yes

4. Is the manuscript presented in an intelligible fashion and written in standard English?

Reviewer #1: Yes

Reviewer #2: Yes

5. Review Comments to the Author

Reviewer #1: The manuscript titled Spatial distribution of suspected and confirmed cholera cases in Mwanza City,

Northern Tanzania by Monica Madullu, MPH

Deborah S.K. Thomas, PhD

Elias C. Nyanza, MPH, PhD

Jeremiah Seni

Sospatro E. Ngallaba, MD

Sophia Kiluvia

Moses Asori

Joseph Kangmennaang,

has been an interesting read. Cholera still remains a health issue in some parts of LMICs. The emphasis on use of GIS for mapping cholera incidence and possible use in making health decisions by government that could reduce cholera in those areas in the manuscript is welcomed.

However, the authors should take note of the following observations;

1. Introduction: Line 54, rephrasing of sentence needed. "Just one decade later in 2015-16, again? more than....."

2. Method: more details on the phase 11 study will be appreciated. For example where all the 852 participants mapped from their residences to the various camps? if so, that means you had access to their home addresses. Did the individuals give consent to this information and use of same?

3. Results: Kindly give more details on distribution of cholera camps across the two districts surveyed. Reconcile details about mortality in line 144-149 where 6 and 7 were mentioned. If they are different, clearly specify.

4. Discussion: Line 192....corroborate studies? or with? line 208, "10% attended more than 24 hours from onset of symptoms" I think you need to check that again. same as line 221-223, it may need rephrasing

Line 229, statement regarding cholera in Northern part near lake Victoria should be added to section where water source closeness in relation to cholera incidence was mentioned (196-200) for better organization. same as line 233, well and tap water user has been mentioned before (203), merge both

line 255-256...rephrase sentence.

5. Reference: check 1,5 and 7. when referencing website, date accessed is usually included

30, add journal or website

6: Table: kindly recheck Table 4, what p-value is considered significant from the analysis you did? this was not mentioned in the methods. from the confidence intervals in the results, p values of 0.10 are not significant with the single asterisk. If you think otherwise, kindly provide more information to the conclusion in method-data analysis section.

& Supporting Information: I successfully downloaded the file and opened it, but could not see data on the file. kindly check again.

Reviewer #2: This paper has the potential to be published because it provides epidemiological insights on the cholera outbreak in a specific region in Tanzania. However, in the introduction I miss why the mapping of the disease was important. Also I am missing strong implications and recommendations that the local authorities can implement to resolve or prevent any future cholera outbreaks.

- Please rewrite the final part of the introduction sentence. I am missing a problem statement that will naturally lead to the aim and the objectives of this research paper and why this research was necessary to be conducted

- Methods: I don’t understand why the wards are mentioned. Please elaborate.

- L84- 92 should be in the introduction section.

- L95 rename to eligibility criterea

- Separate subheading for data analysis

- Please create subheadings to make reading the results easier.

- Please create a separate subheading for limitations and strengths in the discussion section

6. PLOS authors have the option to publish the peer review history of their article (what does this mean?). If published, this will include your full peer review and any attached files.

**Do you want your identity to be public for this peer review?** For information about this choice, including consent withdrawal, please see our Privacy Policy.

Reviewer #1: No

Reviewer #2: No

---

## [Decision Letter · Decision Letter 1]

28 Nov 2022

PGPH-D-22-01133R1

Spatial distribution of suspected and confirmed cholera cases in Mwanza City, Northern Tanzania

Dear Dr. Nyanza,

Thank you for submitting your manuscript to PLOS Global Public Health. After careful consideration, we feel that it has merit but does not fully meet PLOS Global Public Health’s publication criteria as it currently stands. Therefore, we invite you to submit a revised version of the manuscript that addresses the points raised during the review process.

We look forward to receiving your revised manuscript.

Kind regards,

David Musoke, PhD

Academic Editor

Journal Requirements:

Additional Editor Comments (if provided):

May the authors address the minor comment on Table 4.

Reviewers' comments:

Reviewer's Responses to Questions

**Comments to the Author**

1. If the authors have adequately addressed your comments raised in a previous round of review and you feel that this manuscript is now acceptable for publication, you may indicate that here to bypass the “Comments to the Author” section, enter your conflict of interest statement in the “Confidential to Editor” section, and submit your "Accept" recommendation.

Reviewer #1: All comments have been addressed

Reviewer #2: All comments have been addressed

2. Does this manuscript meet PLOS Global Public Health’s publication criteria? Is the manuscript technically sound, and do the data support the conclusions? The manuscript must describe methodologically and ethically rigorous research with conclusions that are appropriately drawn based on the data presented.

Reviewer #1: Yes

Reviewer #2: Yes

3. Has the statistical analysis been performed appropriately and rigorously?

Reviewer #1: Yes

Reviewer #2: Yes

4. Have the authors made all data underlying the findings in their manuscript fully available (please refer to the Data Availability Statement at the start of the manuscript PDF file)?

Reviewer #1: Yes

Reviewer #2: Yes

5. Is the manuscript presented in an intelligible fashion and written in standard English?

Reviewer #1: Yes

Reviewer #2: Yes

6. Review Comments to the Author

Reviewer #1: Authors have effected most observations seen in the original manuscript. However, Table 4 still need review. The notes under the table in the manuscript with track changes and that without track changes are different. the notes with manuscript without tract changes have two asterisks with same interpretation. kindly recheck.

Reviewer #2: All issues were addressed and the content is ready for publication.

7. PLOS authors have the option to publish the peer review history of their article (what does this mean?). If published, this will include your full peer review and any attached files.

**Do you want your identity to be public for this peer review?** For information about this choice, including consent withdrawal, please see our Privacy Policy.

Reviewer #1: No

Reviewer #2: **Yes: **dr. C. Zemouri

---

## [Editor Report · Decision Letter 2]

7 Dec 2022

Spatial distribution of suspected and confirmed cholera cases in Mwanza City, Northern Tanzania

PGPH-D-22-01133R2

Dear Dr. Nyanza,

We are pleased to inform you that your manuscript 'Spatial distribution of suspected and confirmed cholera cases in Mwanza City, Northern Tanzania' has been provisionally accepted for publication in PLOS Global Public Health.

Best regards,

David Musoke, PhD

Academic Editor

Congratulations on the acceptance of your manuscript.